# Microenvironmental modulation breaks intrinsic pH limitations of nanozymes to boost their activities

Tong Li [1,10], Xiaoyu Wang[2,10], Yuting Wang[1], Yihong Zhang[1], Sirong Li[1], Wanling Liu[1], Shujie Liu[1], Yufeng Liu[1], Hang Xing [3], Ken-ichi Otake [4], Susumu Kitagawa [4], Jiangjiexing Wu[1,5] ✉, Hao Dong [2,6,7,8,9] ✉ & Hui Wei [1,6,9] ✉

Functional nanomaterials with enzyme-mimicking activities, termed as nanozymes, have found wide applications in various fields. However, the deviation between the working and optimal pHs of nanozymes has been limiting their practical applications. Here we develop a strategy to modulate the microenvironmental pHs of metal–organic framework (MOF) nanozymes by confining polyacids or polybases (serving as Brønsted acids or bases). The confinement of poly(acrylic acid) (PAA) into the channels of peroxidase-mimicking PCN-222-Fe (PCN = porous coordination network) nanozyme lowers its microenvironmental pH, enabling it to perform its best activity at pH 7.4 and to solve pH mismatch in cascade systems coupled with acid-denatured oxidases. Experimental investigations and molecular dynamics simulations reveal that PAA not only donates protons but also holds protons through the salt bridges between hydroniums and deprotonated carboxyl groups in neutral pH condition. Therefore, the confinement of poly(ethylene imine) increases the microenvironmental pH, leading to the enhanced hydrolase-mimicking activity of MOF nanozymes. This strategy is expected to pave a promising way for designing high-performance nanozymes and nanocatalysts for practical applications.

A wide range of functional nanomaterials has emerged to catalyse enzymatic reactions, which are collectively called nanozymes[1–3]. Considered to be next-generation enzyme mimics, nanozymes have shown promising applications in the fields of bioanalysis[4–6], therapeutics[7,8], environmental protection[9,10], agriculture[11], and so on. To date, they have successfully mimicked various types of enzymes, including peroxidase[12,13], oxidase[14,15], catalase[16,17], superoxide dismutase[18,19], hydrolase[20–22], etc. Like enzymes, the catalytic activities of nanozymes

[1]College of Engineering and Applied Sciences, Nanjing National Laboratory of Microstructures, Jiangsu Key Laboratory of Artificial Functional Materials, Nanjing University, Nanjing, Jiangsu, China. [2]Kuang Yaming Honors School, Nanjing University, Nanjing, Jiangsu, China. [3]Institute of Chemical Biology and Nanomedicine, State Key Laboratory of Chemo/Biosensing and Chemometrics, College of Chemistry and Chemical Engineering, Hunan University, Changsha, Hunan, China. [4]Institute for Integrated Cell-Material Sciences (WPI-iCeMS), Kyoto University, Sakyo-ku, Japan. [5]School of Marine Science and Technology, Tianjin University, Tianjin, China. [6]State Key Laboratory of Analytical Chemistry for Life Science, Nanjing University, Nanjing, Jiangsu, China. [7]Institute for Brain Sciences, Nanjing University, Nanjing, Jiangsu, China. [8]Engineering Research Centre of Protein and Peptide Medicine of Ministry of Education, Nanjing University, Nanjing, Jiangsu, China. [9]Chemistry and Biomedicine Innovation Centre (ChemBIC), ChemBioMed Interdisciplinary Research Centre at Nanjing University, Nanjing, Jiangsu, China. [10]These authors contributed equally: Tong Li, Xiaoyu Wang. ✉e-mail: wujiangjiexing2007@126.com; donghao@nju.edu.cn; weihui@nju.edu.cn

are strongly pH-dependent. Usually, a nanozyme possesses an optimal pH to have the best performance. Nevertheless, a long-standing intrinsic limitation for most nanozymes is that their optimal pHs generally diverge from their operating pHs when employed for practical applications. One representative example is peroxidase-like nanozymes, which have been most extensively studied. While a typical peroxidase-like nanozyme has an optimal pH in the acidic range, its biomedical applications are carried out under the physiological pH (usually neutral pH)[23]. Such a pH deviation has led to the significantly compromised activity and inferior performance of nanozymes. Despite recent efforts devoted to the design of highly active nanozymes[24–28], there still lacks a universal and effective strategy to address the pH limitations. Herein, we propose a microenvironmental modulation strategy to overcome the pH limitations of nanozymes (Fig. 1).

The functional groups of amino acid residues surrounding the active sites provide a unique microenvironment for an enzyme to bind and orient substrates, stabilise transition states, and mediate general acid/base catalysis[29–31]. Inspired by enzymes, we hypothesise that by introducing carboxylic groups (Brønsted acids) to donate protons and amino groups (Brønsted bases) to accept protons, the microenvironmental pH near the catalytic active sites could be decreased and increased, respectively, without changing the pH of bulk solutions. To fulfil our hypothesis, we chose metal−organic frameworks (MOFs) to mimic enzymes because MOFs exhibit features quite analogous to metalloenzymes[32]. On one hand, the metal nodes (or ions) as well as linkers of MOFs can serve as catalytic active sites[10,33]. On the other hand, the cavities and channels in MOFs can be facilely modified, which offers the feasibility to finely tune the microenvironmental pHs. We

reason that polymers with abundant functional groups and high flexibility could easily penetrate into the cavities or channels of MOFs and then be confined near the catalytic active centres to locally modulate the pHs[34–37].

In this study, we demonstrate that the confinement of poly(acrylic acid) (PAA) and poly(ethylene imine) (PEI) within MOF nanozymes effectively increases the local concentrations of protons and hydroxyl ions, leading to a decrease and an increase in the local pHs, respectively (Fig. 1). The modulated pHs, in turn, boost the peroxidase-like and hydrolase-like activities of the MOF nanozymes. We speculate that the microenvironmental modulation strategy can not only break the pH limitations of current nanozymes and therefore broaden the application of nanozymes but also pave a promising way for the modulation of other microenvironmental properties of nanocatalysts.

## Results and discussion
### Peroxidase-like activity modulation through poly(acrylic acid) confinement

Most of the reported nanozymes are peroxidase mimics that catalyse the peroxidation of a substrate such as 3,3′,5,5′-tetramethylbenzidine dihydrochloride (TMB) in the presence of $H_2O_2$. Therefore, we investigated the microenvironmental pH modulation of peroxidase-like nanozymes. Current peroxidase-like nanozymes, including MOF ones, have their optimal activities under acidic pHs, as evidenced by our literature survey. As shown in Fig. 2a, over the last three years, most of the reported optimal pHs lay in the range from pH 2.5 to pH 4.5. This might be because the protons participate in the peroxidation, i.e., $2TMB+H_2O_2+2H^+ \rightarrow 2TMB^{\bullet+}+2H_2O$. Besides, TMB is prone to be oxidised at acidic pHs (Supplementary Figs. 1 and 2). Since peroxidase-like

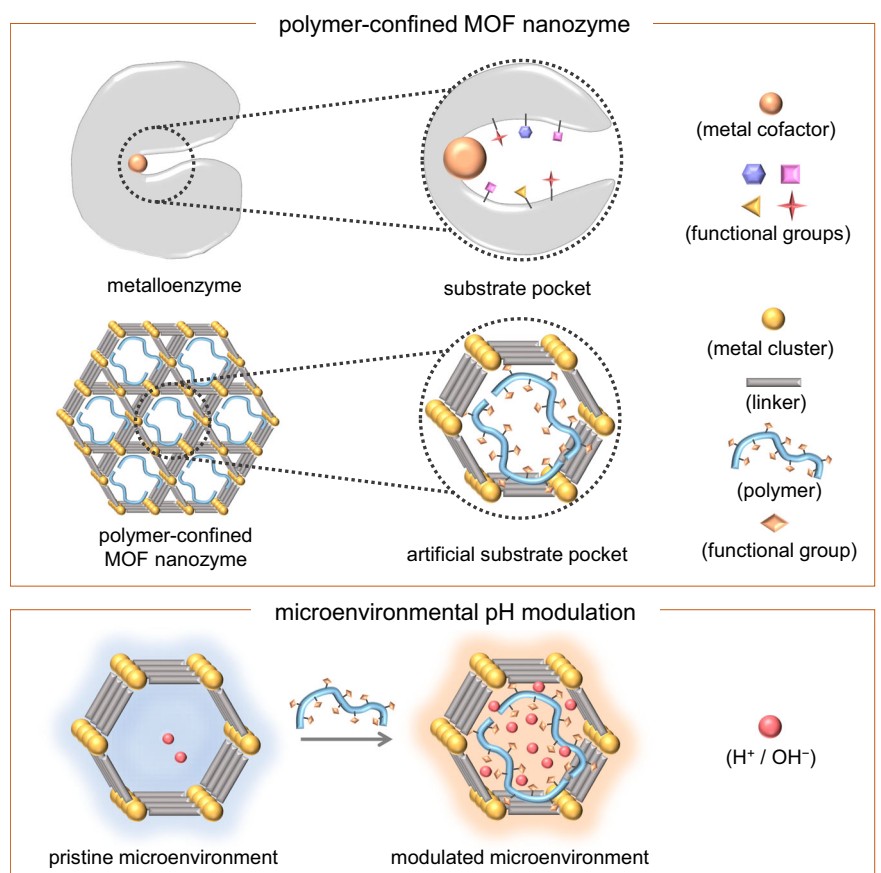

**Fig. 1 | Breaking pH limitations of MOF nanozymes to boost their activities.** Upper: Schematic illustration of microenvironments surrounding active sites in a metalloenzyme and a polymer-confined MOF nanozyme. Lower: Confined

polymers modulate the microenvironmental pHs of MOF nanozymes through proton donation and acceptance.

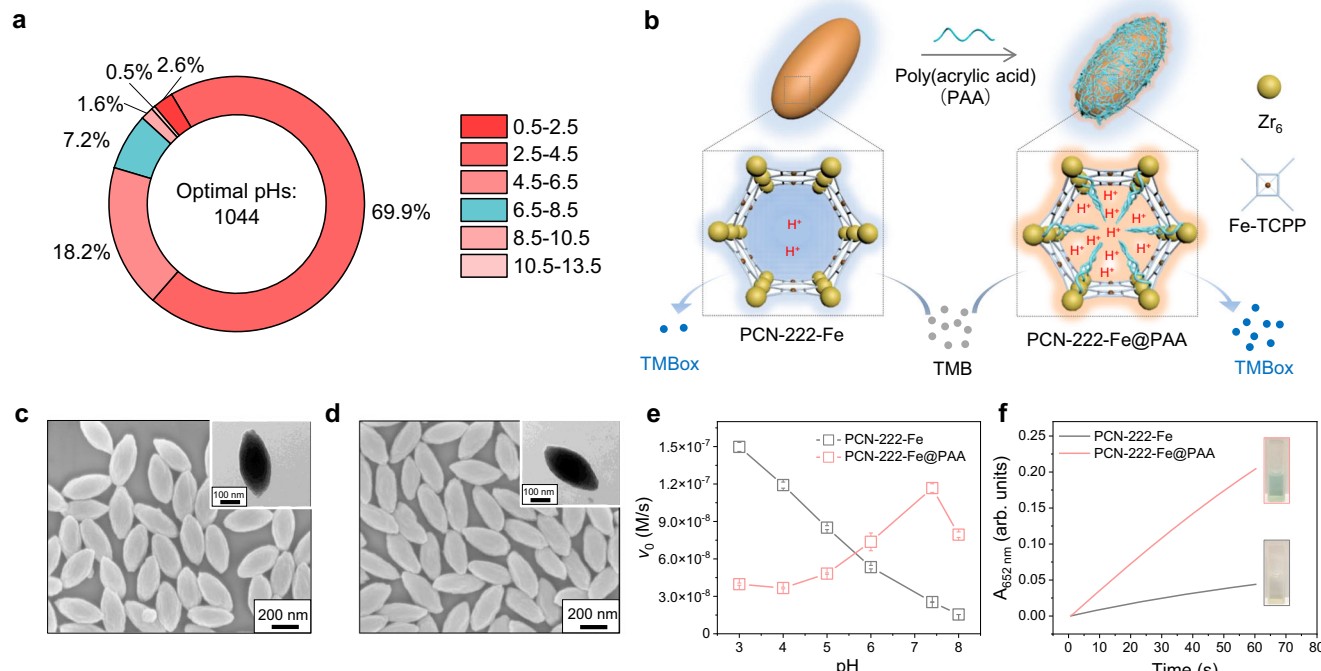

**Fig. 2 | Engineering physiological pH active peroxidase-like nanozymes through poly(acrylic acid) confinement. a** Optimal pHs of peroxidase-like nanozymes reported from 2019 to 2021. **b** Scheme of confining PAA within PCN-222-Fe NPs to form PCN-222-Fe@PAA NPs, leading to an increase in protons near the catalytic active sites and subsequent improvement in the activity of MOF nanozymes. For clarity, the enlarged models were not drawn to scale. SEM images and inset TEM images of (**c**) PCN-222-Fe NPs and (**d**) PCN-222-Fe@PAA NPs taken from one batch of NPs. **e** pH-dependent initial velocity ($v_0$) of catalytic peroxidation of TMB, showing the influence of pH on the peroxidase-like activity of PCN-222-Fe NPs and PCN-222-Fe@PAA NPs. **f** Time evolution of absorbance at 652 nm ($A_{652\,nm}$) for monitoring the peroxidase-mimicking catalytic activities of 5 μg/mL PCN-222-Fe NPs or PCN-222-Fe@PAA NPs at R.T., under the condition of 200 mM Tris buffer (pH 7.4) containing 0.2 mM TMB and 0.2 mM $H_2O_2$. The inset photos show the colour of the reaction solutions. Data in (**e**) are expressed as mean values ± SEM, $n = 3$.

nanozymes have been mainly used under physiological conditions where the pH ranges from 6.5 to 7.4, it is necessary to break the intrinsic pH limitations to boost their catalytic activity and application performance. Herein, the peroxidase-like PCN-222-Fe (PCN = porous coordination network) nanoparticle (NP), constructed by $Zr_6$ clusters and Fe-TCPP (TCPP = tetrakis(4-carboxyphenyl)porphyrin), was employed as a model MOF nanozyme. The linker, Fe-TCPP, serves as the catalytic active site. Poly(acrylic acid) with an $M_w$ of 2 kDa (designated as PAA) was used to modify the mesoporous channels of PCN-222-Fe NPs. The abundant carboxyl groups of PAA confined within PCN-222-Fe NPs would increase the local proton concentration and subsequently improve the catalytic peroxidation of TMB (Fig. 2b).

PCN-222-Fe NPs were fabricated with minor modifications as reported[38]. As shown in Fig. 2c, the obtained PCN-222-Fe NPs exhibit a spindle-like shape, which is consistent with previous studies[39]. The length and width of PCN-222-Fe NPs are 360 ± 35 nm and 171 ± 16 nm, respectively (Supplementary Fig. 3). The XRD pattern of NPs was also measured, which corresponds well with that of the simulated PCN-222 (Supplementary Fig. 4). After incubation with PAA, Fe-TCPP in PCN-222-Fe NPs was barely released (Supplementary Fig. 5). The obtained PAA-modified PCN-222-Fe NPs are termed as PCN-222-Fe@PAA NPs (Fig. 2d). The length and width of PCN-222-Fe@PAA are 368 ± 30 nm and 166 ± 16 nm, respectively (Supplementary Fig. 3). A zeta potential reversal from about +20 mV of PCN-222-Fe NPs to −25 mV of PCN-222-Fe@PAA NPs was observed, indicating the successful modification of PAA (Supplementary Fig. 6). The decreased nitrogen uptake and pore volume of PCN-222@PAA NPs compared with PCN-222 NPs indicates that PAA could be confined within the MOF channels (Supplementary Fig. 9). Furthermore, the binding energy of Zr $3d_{5/2}$ increases from 182.4 eV to 182.6 eV after PAA modification (Supplementary Fig. 10),

which proves the modification of PAA via the coordination between $Zr_6$ clusters of PCN-222-Fe NPs and carboxyl groups of PAA[40]. In addition, the amount of PAA in PCN-222-Fe@PAA NPs was measured to be about 6.6% according to the thermal gravimetric (TG) analysis (Supplementary Fig. 11). Combined, these results demonstrate the successful confinement of PAA within PCN-222-Fe NPs to form PCN-222-Fe@PAA NPs.

We then investigated the influence of pH on the peroxidase-like activities of PCN-222-Fe NPs and PCN-222-Fe@PAA NPs in the presence of $H_2O_2$ and TMB. As shown in Fig. 2e, the pristine PCN-222-Fe NPs exhibit the highest activity at pH 3.0, and the activity decreases with the increase of pH from 3.0 to 8.0. The activity of PCN-222-Fe NPs at pH 7.4 is around 17% of that at pH 3.0. In contrast, PCN-222-Fe@PAA NPs exhibit the highest activity at pH 7.4. A further comparison between the activity of PCN-222-Fe@PAA NPs and that of PCN-222-Fe NPs was carried out at pH 7.4. The result shows that the activity of PCN-222-Fe NPs increases by around 4 folds after modification with PAA, which is also distinctly indicated by the colour differences of TMB solutions after 1 min of reaction (Fig. 2f). Besides, PCN-222-Fe@PAA NPs exhibit a good pH-dependent operational stability (Supplementary Fig. 12). To identify whether the modulation strategy can improve the peroxidase-like activity of PCN-222-Fe NPs for other substrates, a classic fluorescent substrate, Amplex Red, was studied. A 5-fold increase in the initial velocity of Amplex Red peroxidation was achieved through PAA confinement (Supplementary Fig. 13). PAA is a pH-responsive polymer which shrinks under acidic condition[41]. As the organic linker, Fe-TCPP serves as the catalytic active site, the activity decrease of PCN-222-Fe NPs at acidic pHs after PAA confinement could be ascribed to obstruction of catalytic active sites by shrinking PAA chains (Supplementary Figs. 14–16).

**PAA confinement decreases the microenvironmental pH of MOF nanozymes to boost their peroxidase-like activity at neutral pH**

We carried out a series of experiments to study the mechanism of peroxidase-like activity increase at neutral pH through PAA confinement. First, we investigated the potential effects of free PAA, the affinity of PCN-222-Fe@PAA NPs towards the substrate $H_2O_2$, and the electrostatic interaction between PCN-222-Fe@PAA NPs and the substrate TMB. As shown in Supplementary Figs. 17–20, none of these factors could significantly increase the peroxidase-like activity of PCN-222-Fe@PAA NPs.

We then probed whether the PAA confinement could lower the microenvironmental pH of PCN-222-Fe NPs. It is known that the porphyrins are responsive to pH change because of the protonation of their nitrogen atoms, which leads to the shift of their absorption peaks[42]. However, an iron atom located in the centre of a TCPP molecule inhibits the protonation of nitrogen atoms. Hence, we employed TCPP instead of Fe-TCPP to construct PCN-222 NPs, which have an identical crystal structure to PCN-222-Fe NPs. We would be able to probe the local pH change by monitoring the light absorption and the corresponding colour of PCN-222 NPs after different treatments (Fig. 3a). The obtained PCN-222 suspension is purple, as shown in Fig. 3b. Once it is incubated with PAA for 30 min (i.e., PAA modification procedure), the colour of the suspension changes to green. The same colour change from purple to green also occurs with the addition of HCl. Interestingly, the PAA-modified PCN-222 suspension, termed PCN-222@PAA, remains green after several washings with water, while the PCN-222 suspension incubated with HCl returns to purple after the same washing. We then redispersed PCN-222@PAA NPs in 0.2 M Tris buffer with a pH of 7.4. The colour of PCN-222@PAA NPs changes from green to brown rather than back to purple, which demonstrates that some protons remain within MOF NPs in Tris buffer at pH 7.4. This phenomenon indicates that PAA modification could increase the local proton concentration inside PCN-222 NPs. Besides, the SEM images of PCN-222 NPs and PCN-222@PAA NPs in Fig. 3c, d demonstrate their basically identical morphology and size. The XRD pattern of PCN-222 NPs remains

unchanged after PAA modification (Supplementary Fig. 21). The zeta potential decrease proves the successful modification of PAA (Supplementary Fig. 22). To deepen our insight into the colour change, we monitored absorption spectra of PCN-222 NPs with the addition of protons. As exhibited in Supplementary Fig. 23a, the PCN-222 suspension has a characteristic absorption peak at around 420 nm. With the addition of HCl, a new absorption peak at around 445 nm appears with the disappearance of the peak at 420 nm. The inset photos also show that the colour of the PCN-222 suspension changes from light purple to green with the addition of HCl. The ratio of $A_{445\,nm}/A_{420\,nm}$ increases when the protons amount increases, which is shown in Supplementary Fig. 23b. We then monitored absorption spectra of PCN-222 NPs and PCN-222@PAA NPs in water, 10 mM Tris buffer, and 200 mM Tris buffer. The pH of Tris buffers is 7.4. It is shown that $A_{445\,nm}/A_{420\,nm}$ values of PCN-222@PAA NPs are higher than those of PCN-222 NPs in both water and Tris buffers, which proves the retainment of protons within PCN-222@PAA NPs in neutral buffer solutions. Besides, the $A_{445\,nm}/A_{420\,nm}$ value of PCN-222@PAA NPs decreases with increase of Tris buffer concentration, which can be ascribed to that some protons are buffered by Tris molecules (Supplementary Fig. 24 and Fig. 3e). To prove that the absorption response of PCN-222 towards protons comes from the linker, TCPP, we measured the Vis absorption spectra of TCPP DMF solution with the addition of HCl. As shown in Supplementary Fig. 25, an identical change in the absorption peaks occurs to that of the PCN-222 suspension.

The fluorescence emission of TCPP is also inherently responsive to the change of pH, which endows TCPP-based MOFs such as PCN-225 with pH-sensitive fluorescence[43]. As shown in Supplementary Fig. 26, the fluorescence intensity of PCN-222 NPs is lowered by decreasing the pH of solutions, which corresponds well with a previous study[43]. Therefore, we recorded the fluorescence emission spectra of PCN-222 NPs and PCN-222@PAA NPs to further investigate the decrease of microenvironmental pH after PAA modification (Supplementary Fig. 27 and Fig. 3f). PCN-222@PAA NPs exhibit a much lower fluorescence intensity than PCN-222 NPs in water.

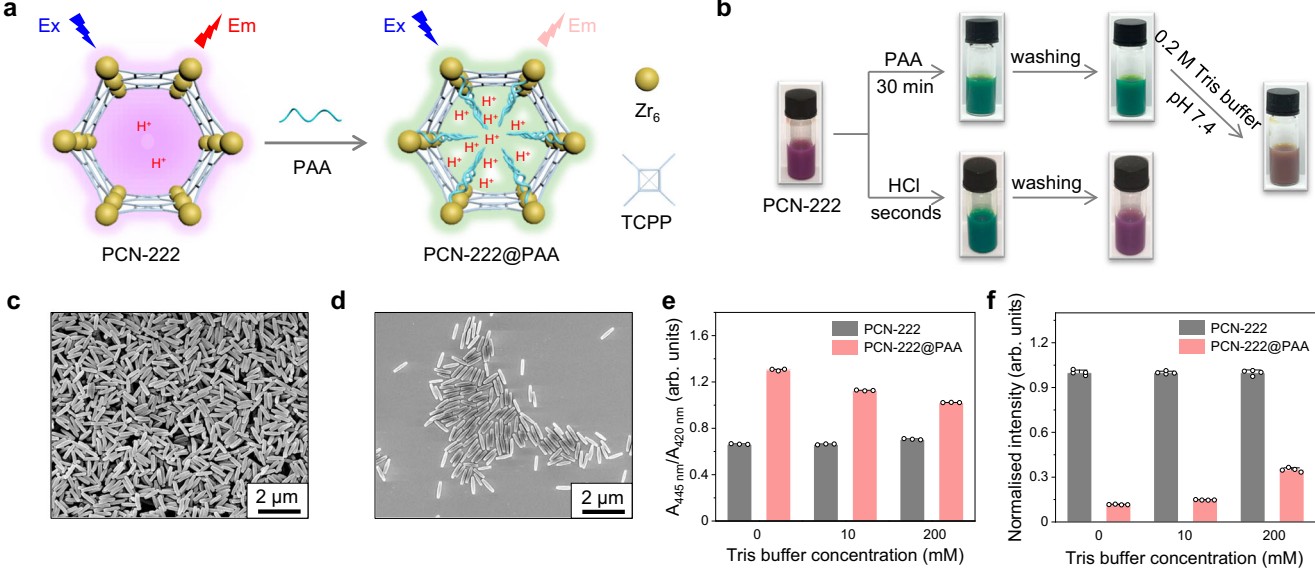

**Fig. 3 | Proof of microenvironmental pH decrease through PAA confinement.**
**a** Scheme of PCN-222 NPs and PCN-222@PAA NPs. The light absorption and fluorescence of PCN-222 NPs are intrinsically responsive to proton concentrations. **b** Photos of 0.5 mg/mL PCN-222 NPs through PAA modification or introduction of HCl. **c, d** SEM images of PCN-222 NPs and PCN-222@PAA NPs, respectively, taken from one batch of NPs. **e** Ratios of absorbance at 445 nm to absorbance at 420 nm

($A_{445\,nm}/A_{420\,nm}$) of 10 μg/mL PCN-222 NPs and PCN-222@PAA NPs in Tris buffer solutions with different concentrations at pH 7.4. **f** Normalised fluorescence intensities at 660 nm of 20 μg/mL PCN-222 NPs and PCN-222@PAA NPs in Tris buffer solutions with different concentrations at pH 7.4. $\lambda_{ex} = 420$ nm. Data in (**e**) are expressed as mean values ± SEM, $n = 3$. Data in (**f**) are expressed as mean values ± SEM, $n = 4$. Data are expressed as mean ± standard error of 3 experiments.

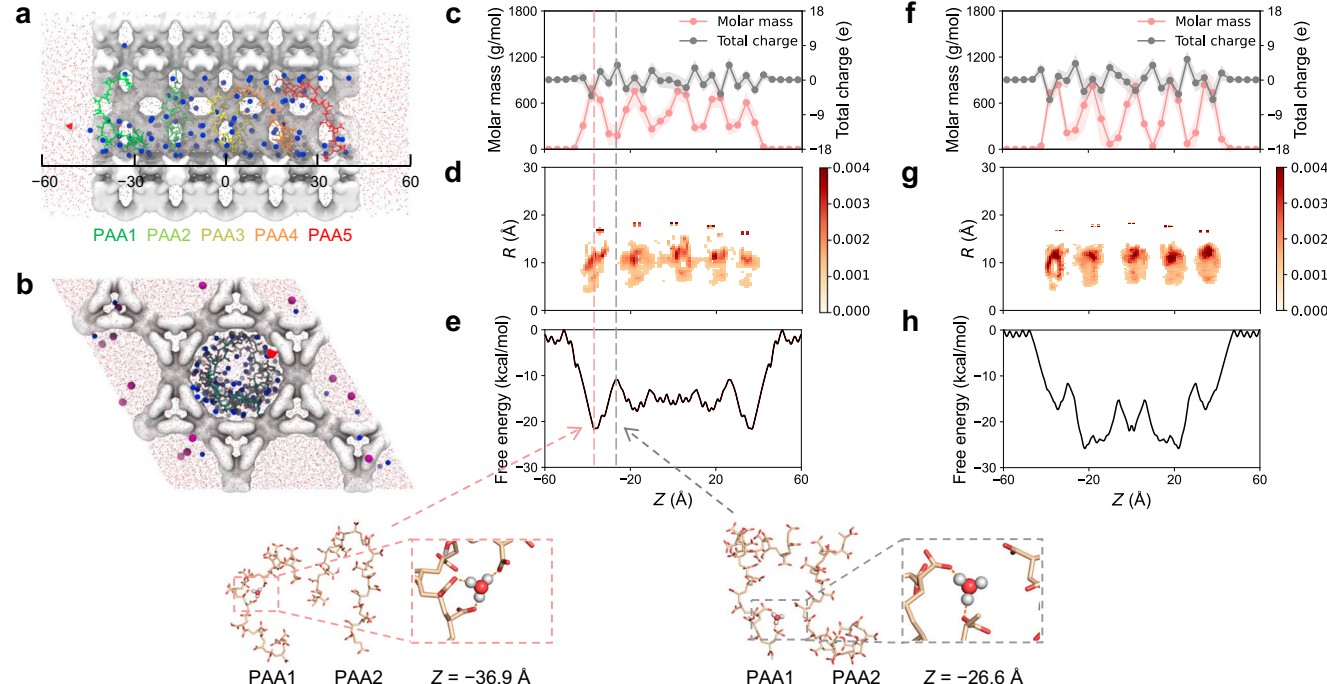

**Fig. 4 | MD simulations identify the regulatory mechanism of microenvironmental pH in a PAA-modified channel of PCN-222-Fe@PAA. a** Cross-sectional and (**b**) side views of the system. The centre of mass of the MOF was placed at (0, 0, 0), and the channel was aligned along the $Z$-axis. The $Z$-direction length of the whole system is 120 Å, of which −43 Å-43 Å is the range of MOF, and the rest is the water layer with a thickness of -34 Å. The five PAA chains are in different colours. The blue and purple spheres represent the nitrogen atom of the Tris group and the chloride ions. Hydronium is shown using the space-filling model.

**c** Distribution of charge and mass of PAA in the 22/5 system. The light pink and grey backgrounds represent the error bands for molar mass and total charge, respectively. **d** 22/5 PAA chains show a relatively narrow distribution within the confined environment with less overlap between chains. **e** Permeation free energy of hydronium through the pore. The shape of the curve is highly associated with the population of PAA. Two snapshots show representative binding modes of hydronium with PAA. The information for the 27/0 system is shown in (**f–h**).

When Tris buffer (pH = 7.4) with increasing concentrations from 10 mM to 200 mM is employed, PCN-222@PAA NPs still show a lower fluorescence intensity than PCN-222 NPs, although in a less degree: water >10 mM Tris >200 mM Tris (of note, 200 mM Tris buffer is applied as the reaction medium in the activity measurement, Fig. 2f). These results indicate that PAA modification lowers the microenvironmental pH near the catalytically active sites of PCN-222-Fe NPs, leading to the improved activity under neutral conditions. Indeed, we proved that PAA modification on the surface of MOF NPs could not effectively lower the microenvironmental pH. PAA within the pores of MOF NPs is able to decrease local pH thus achieving the activity improvement (Supplementary Figs. 28 and 29). Besides, PAA confinement within MOF NPs is quite stable due to the coordination interaction between carboxyl groups of PAA and $Zr_6$ clusters of MOF NPs (Supplementary Fig. 30). Other polymers, such as poly(sodium 4-styrenesulfonate) (PSS), poly(allylamine hydrochloride) (PAH), and poly(ethylene imine) (PEI), were also employed to modulate the activity of PCN-222-Fe NPs. As shown in Supplementary Fig. 31, neither PSS, PAH nor PEI significantly influences the activity. Besides, it is indicated that deprotonated PAA is not able to enhance the peroxidase-like activity of PCN-222-Fe NPs, which emphasises the essence of protons donation by PAA (Supplementary Fig. 32). For horseradish peroxidase when $H_2O_2$ molecules are adsorbed on the iron centre, protons help activate heterolytic O−O bond cleavage forming highly oxidising ferryl species with the assistance of surrounding functional groups[44]. For PCN-222-Fe NPs, it is proposed that microenvironmental accumulation of protons also helps facilitate O−O heterolytic cleavage leading to the formation of highly oxidising ferryl species under neutral conditions.

## Molecular dynamics simulations reveal the mechanism of microenvironmental pH regulation

We then performed molecular dynamics (MD) simulations to reveal the mechanism of microenvironmental pH regulation. PAA is a weak acid with an apparent $pK_a$ value determined to be -6.07 (Supplementary Fig. 33). Different from strong acids, confined PAA chains cannot release protons entirely in neutral solutions. It might be because deprotonated carboxyl groups in PAA is able to hold protons within the channels in MOFs. To prove this, the 22/5 (22 carboxyl groups on a PAA chain are deprotonated and the other 5 are in protonated states, representing a weakly acidic system) and 27/0 (all 27 carboxyl groups on a PAA chain are deprotonated, representing a neutral or slightly basic system) were used for MD simulations. The protonation state of PAA varies in response to the change in pH of the bulk phase, thus modulating the microenvironmental pH. For the 22/5 system, MD simulations (Fig. 4 and Supplementary Fig. 34) show that the PAA chains adhere to the inner side of the pore and adopt curled shapes (Fig. 4a), which significantly reduces the pore diameter but does not absolutely block it (Fig. 4b). Meanwhile, each PAA chain is localised along the pore axis, most likely due to electrostatic repulsion with adjacent chains (Fig. 4c, d). Such structural features provide a unique confinement environment for protons. The free energy profile of hydronium permeation through the 22/5 MOF channel shows that protons are prone to be held by the confined lumen (Fig. 4e). In brief, the free energy is lower where PAA is more distributed. Specifically, starting from the bulk phase, the free energy decreases significantly as the hydronium gradually approaches and enters the MOF until it is -22 kcal/mol lower than that in the bulk. This is the position where the mass distribution of the outermost PAA chain reaches the maximum ($Z = −36.9$ Å). The hydronium is now stabilised by three negatively

charged carboxyl groups on PAA and a durable hydrogen bonding with a pore-water. After that, the free energy gradually increases until the mass distribution of PAA reaches a minimum ($Z = -26.6$ Å), although it is still ~10 kcal/mol lower than that in the bulk. Now, the hydronium is located between the outermost PAA chain and a neighbouring chain. Notably, ab initio calculations suggest that the binding strength of PAA to the hydronium, the Tris group, and the reaction product oxTMB decreases sequentially (Supplementary Table 2). Therefore, PAA enriches the proton within the channel without hindering the release of oxTMB.

The 27/0 also exhibits the capacity to capture protons (Fig. 4f–h). The lowest free energy of hydronium in the 27/0 system is slightly lower than that in 22/5, indicating that 27/0 is more likely to hold protons in the channel and thus could exert higher catalytic activity. Notably, the simulations are consistent with our experimental observation that the catalytic activity of PCN-222-Fe@PAA increased continuously with increasing pH and reached the highest activity at pH = 7.4, after which the activity decreased (Fig. 2e). The decrease in activity under basic conditions is probably because the high concentration of hydroxyl groups offsets the enrichment effect of channels for protons.

Seemingly, PAA acts as a buffer to modulate the microenvironmental pH: it counteracts the increase/decrease in the local [H⁺] within the pore by uptaking/releasing protons. This delicate regulation mechanism effectively shifts the pH range where PCN-222-Fe works, enabling it to efficiently catalyse reactions in a neutral pH solution. Nevertheless, the highest activity of PCN-222-Fe@PAA is slightly lower than that of PCN-222-Fe (Fig. 2e), indicating that the diffusion of reactants/products in the confined pore may partially attenuate the efficiency of the reaction.

## Activity comparison between PCN-222-Fe@PAA NPs and other peroxidase-like nanozymes

Considering the substantial activity improvement of PCN-222-Fe@PAA NPs at physiological pH via microenvironmental pH modulation, we compared their peroxidase-like activity with other typical peroxidase-like nanozymes, including $Fe_3O_4$ NPs, graphene oxide, $NH_2$-MIL-88B, and Pt NPs. The successful synthesis of these four nanozymes was confirmed by TEM imaging and hydrodynamic size measurements (Supplementary Fig. 35). Encouragingly, PCN-222-Fe@PAA NPs exhibit much higher activity than the four nanozymes at pH 7.4, adopting an identical mass concentration (Supplementary Fig. 36). These results confirm the good performance of PCN-222-Fe@PAA NPs at physiological pH, demonstrating the power of microenvironmental modulation for engineering the activity of MOF nanozymes.

## Solving pH mismatch between natural oxidases and MOF nanozymes through microenvironmental pH modulation

Enzymatic cascade systems catalysed by natural oxidases coupled with peroxidase-mimicking nanozymes have been widely employed in the fields of bioanalysis, diagnosis, and therapeutics[45,46]. The produced $H_2O_2$ catalysed by oxidases serves as the substrate of peroxidase-mimicking nanozymes accompanied by peroxidation of a colorimetric or fluorescent substrate. However, there has been a persistent pH mismatch between oxidases and peroxidase-mimicking nanozymes which hinders the development of the cascade systems. Most oxidases exhibit their best activities under neutral conditions, while the optimal pHs of most peroxidase-mimicking nanozymes are acidic. Although previous studies improve the activity of peroxidase-mimicking nanozymes, such a pH mismatch has not been well addressed[24–26,28]. Here, we shifted the optimal pH of peroxidase-mimicking nanozymes from acidic to neutral, which effectively overcame the pH mismatch between oxidases and peroxidase-mimicking nanozymes (Fig. 5a).

Due to the pH mismatch, people usually perform the oxidase-catalysed reaction first under neutral conditions, and then the reaction solution is transferred into an acidic media for subsequent reaction catalysed by peroxidase-mimicking nanozymes[47]. Such a two-step procedure is generally time-consuming and inefficient. First, the produced intermediate $H_2O_2$ in the oxidase-catalysed step cannot directly engage in the nanozyme-catalysed step. Second, when the produced $H_2O_2$ is added to the acidic medium in the nanozyme-catalysed step, it gets diluted, thereby lowering the activity of nanozymes. In our study, we perform one-pot oxidase/nanozyme cascade reactions directly at pH 7.4 through microenvironmental pH modulation of nanozymes. Both oxidases and nanozymes could function in their optimal activities, and the produced intermediate $H_2O_2$ could be efficiently consumed by nanozymes without accumulation (Fig. 5b).

PCN-222-Fe nanozymes exhibit quite higher activity at pH 4.0 than pH 7.4, and a significant activity improvement is achieved through PAA confinement at pH 7.4 (Supplementary Fig. 37). We then monitored the catalytic activity of four natural oxidases including urate oxidase (UOx), lactate oxidase (LOx), choline oxidase (COx), and alcohol oxidase (AOx) at pHs 4.0 and 7.4. It is shown that the four oxidases exhibit a much lower activity at pH 4.0 than pH 7.4 (Supplementary Figs. 38 and 39). Due to the pH mismatch between the four oxidases and PCN-222-Fe NPs, oxidase/PCN-222-Fe cascade reactions exhibit an inferior efficiency at both pH 4.0 and pH 7.4. Through PAA modification, the operating pH of the four oxidases matches well with the optimal pH of PCN-222-Fe@PAA NPs, leading to a significant enhancement in the cascade efficiency under neutral conditions (Supplementary Figs. 38 and 39). Due to the good substrate specificity of oxidases, the one-pot oxidase/MOF nanozyme cascade reaction serves as an effective toolbox for the specific detection of target biomolecules. We detected four biomolecules including uric acid, lactic acid, choline, and ethanol enabled by oxidase/PCN-222-Fe@PAA cascade reactions, which showed relatively low limits of detection (LOD) (Supplementary Fig. 40a, c, e, g). Besides, oxidase/PCN-222-Fe@PAA cascade reactions exhibit good sensing specificity towards those target biomolecules (Supplementary Fig. 40b, d, f, h).

Besides the four oxidases, we also employed another oxidase, D-amino acid oxidase (DAAO), which specifically catalyses the oxidation of D-amino acids rather than L-amino acids. Activity measurement shows that DAAO exhibits a much lower activity at pH 4.0 than pH 7.4 (Supplementary Fig. 41 and Fig. 5c, d). Here, we constructed a DAAO/PCN-222-Fe@PAA cascade system which was well performed for chirality recognition of amino acids in a one-pot manner (Fig. 5e). We first employed alanine as the substrate. It is demonstrated that the introduction of D-alanine rather than L-alanine initiates the cascade reaction catalysed through DAAO/PCN-222-Fe@PAA system (Fig. 5f). Besides alanine, we also achieved the chirality recognition of other three amino acids including valine, proline, and isoleucine. The addition of interferences does not influence the chirality recognition, which proves the anti-interference ability of the cascade system (Fig. 5g). Moreover, chirality recognition of amino acids was also achieved in a fluorescent manner yielding a quite high signal intensity (Fig. 5h). Besides natural amino acids, the chirality of four non-natural amino acids was also recognised (Fig. 5i). We speculate that microenvironmental pH modulation of nanozymes will broaden our insight into construction of highly efficient oxidase/nanozyme cascade systems, which will find wide applications in various fields.

## Modulating hydrolase-like activity of MOFs through poly(ethylene imine) confinement induced microenvironmental pH increase

We have achieved a decrease in microenvironmental pH through PAA confinement. To explore the breadth of our microenvironmental pH

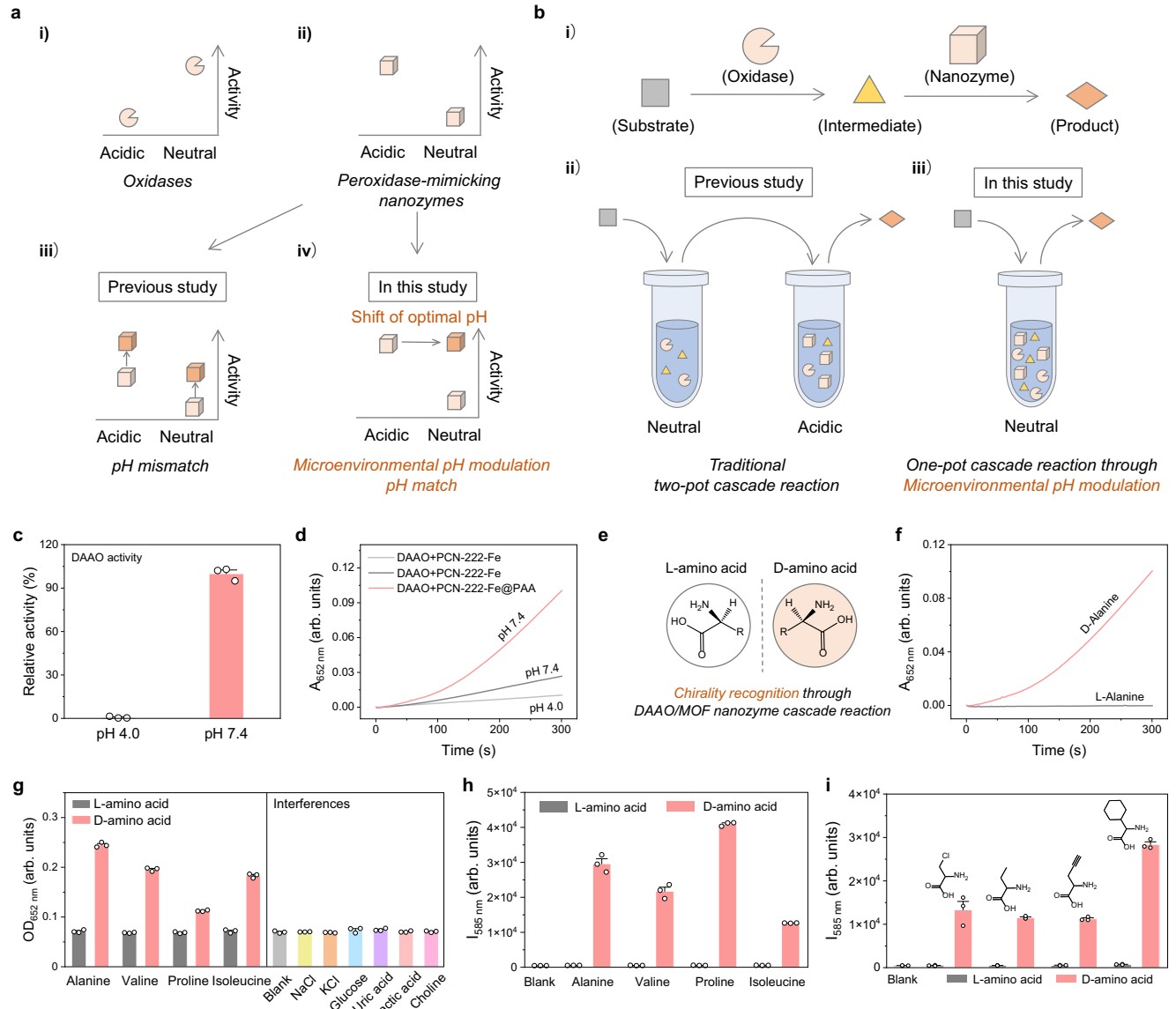

**Fig. 5 | One-pot oxidase/MOF nanozyme cascade reactions enabled by microenvironmental pH modulation. a** Schematic illustration of pH mismatch between oxidases and peroxidase-mimicking nanozymes, which can be overcome through microenvironmental pH modulation. **b** Schematic illustration of a one-pot cascade reaction catalysed by an oxidase coupled with a peroxidase-mimicking nanozyme, which can be fulfilled through microenvironmental pH modulation. **c** Relative activity of D-amino acid oxidase (DAAO) at pHs 4.0 and 7.4. **d** Time evolution of absorbance at 652 nm ($A_{652\,nm}$) for monitoring one-pot cascade reactions catalysed by DAAO coupled with PCN-222-Fe NPs at pHs 4.0 and 7.4, or PCN-222-Fe@PAA NPs at pH 7.4. **e** Schematic illustration of L-amino acid and D-amino acid, which can be recognised through DAAO/MOF nanozyme cascade reaction. **f** Time evolution of $A_{652\,nm}$ for chirality recognition of alanine through DAAO/PCN-222-Fe@PAA cascade reaction. **g** Optical density at 652 nm ($OD_{652\,nm}$) for chirality recognition of four natural amino acids including alanine, valine, proline, and isoleucine using a microplate reader through DAAO/PCN-222-Fe@PAA cascade reaction. **h** Fluorescence intensity at 585 nm ($I_{585\,nm}$) for chirality recognition of natural amino acids as mentioned in (**g**). $\lambda_{ex} = 560$ nm. **i** $I_{585\,nm}$ for chirality recognition of four non-natural amino acids. Data in (**c**, **g**, **h**, and **i**) are expressed as mean values ± SEM, $n = 3$.

modulation strategy, we speculate that the confinement of polybases within MOFs would increase the microenvironmental pH. Here, we used poly(ethylene imine) (PEI) with abundant amine groups as a model polybase to increase the microenvironmental pH of zirconium-based MOF (Zr-MOF) nanozymes and boost their hydrolase-like activities. For Zr-MOFs, the $Zr_6$ clusters can serve as the catalytic active centres. We incubated Zr-MOF nanozymes with PEI and further employed poly(ethylene glycol) diglycidyl ether (PEGDE) to crosslink the confined PEI through the reaction between the primary amino groups of PEI and the epoxide groups of PEGDE (Fig. 6a and Supplementary Fig. 42). A typical substrate, p-nitrophenyl phosphate (pNPP), was employed to analyse the hydrolase-like activity of Zr-MOF nanozymes. The product, p-nitrophenyl (pNP), can be monitored based on

its characteristic absorption. MOF-808 NPs possess 1.8 nm pores, which enable the substrates to access metal sites inside NPs. Besides, $Zr_6$ nodes in MOF-808 NPs are six-connected, which expose more catalytically active sites than other Zr-MOF NPs including 12-connected UiO-66 NPs and 8-connected NU-1000 NPs[48,49]. Hence, we fabricated MOF-808 NPs as our study mode, which were imaged in Fig. 6b. After PEI incubation and PEGDE crosslinking, the obtained NPs are denoted as MOF-808@PEI/PEGDE NPs (Fig. 6c). Furthermore, MOF-808@PEI/PEGDE NPs exhibit decreased nitrogen uptake and pore size compared with MOF-808 NPs, which indicates the successful confinement of PEI within the cavities of MOF NPs (Supplementary Fig. 43). The TG result proves that the PEI amount coupled with PEGDE is around 2.9% (Supplementary Fig. 44). After optimising the PEI confinement procedure

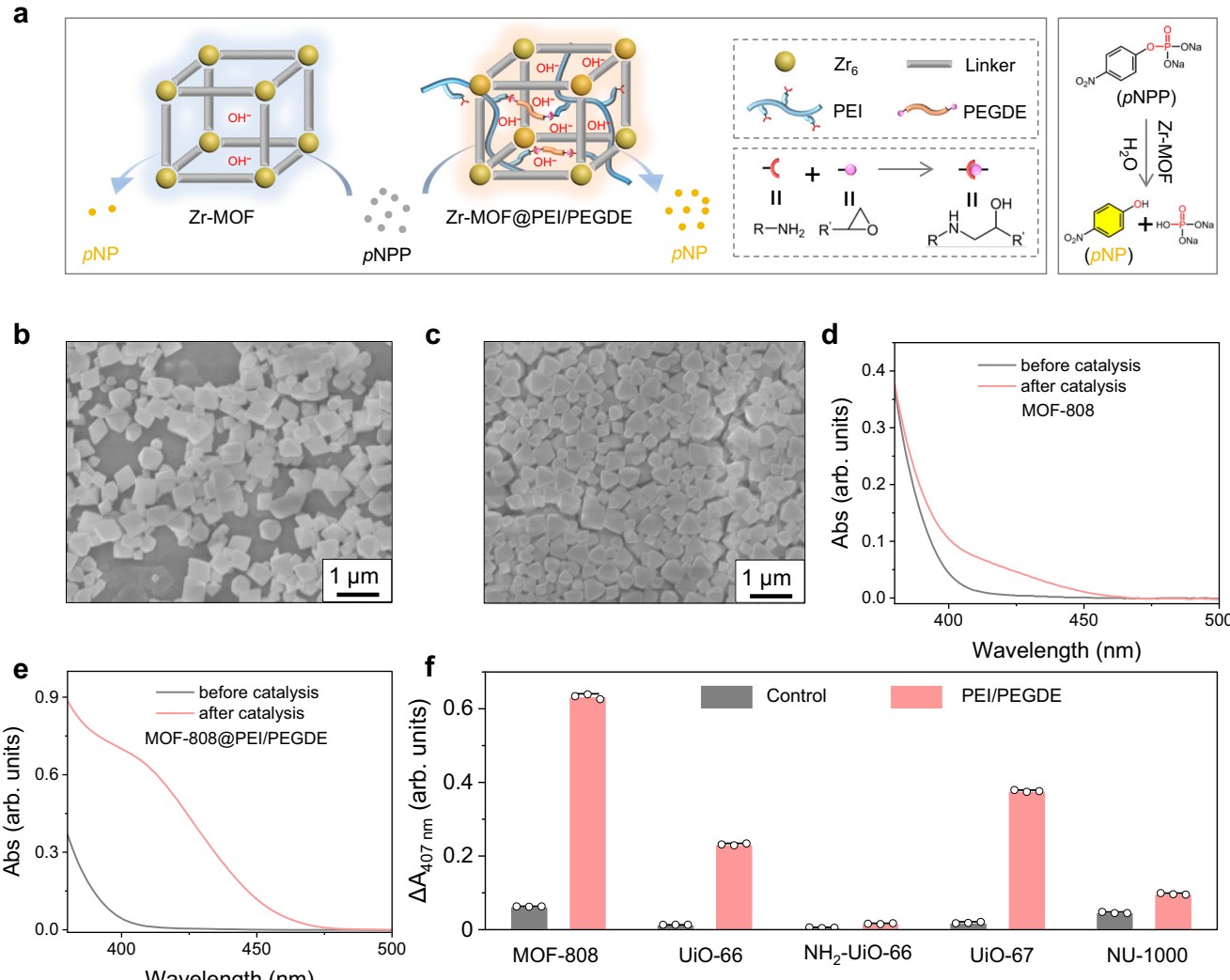

**Fig. 6 | Microenvironmental pH increase of Zr-MOFs through poly(ethylene imine) confinement for hydrolytic activity improvement. a** Scheme of structures of Zr-MOF and Zr-MOF@PEI/PEGDE with increased microenvironmental hydroxide ions, and enhanced catalytic hydrolysis of substrate pNPP. SEM images of (**b**) MOF-808 NPs and (**c**) MOF-808@PEI/PEGDE NPs taken from one batch of NPs. UV–vis spectra of pNPP solutions before and after 20-min incubation with (**d**) MOF-808 NPs, and (**e**) MOF-808@PEI/PEGDE NPs. **f** Absorbance changes at 407 nm ($\Delta A_{407\,nm}$) of pNPP before and after 20-min incubation with Zr-MOF NPs and Zr-MOF@PEI/PEGDE NPs including MOF-808 NPs, UiO-66 NPs, NH2-UiO-66 NPs, UiO-67 NPs, and NU-1000 NPs. Data in (**f**) are expressed as mean values ± SEM, $n = 3$.

(Supplementary Figs. 45–47), we used MOF-808@PEI/PEGDE NPs to investigate whether the PEI confinement would increase the hydrolytic activity of MOF-808 nanozymes. As shown in Fig. 6d, e, the confinement of PEI increases the hydrolytic activity of MOF-808 nanozyme by around 9 folds. To exclude the potential effect of PEI-induced increase of bulk pH on the activity improvement, a basic pH indicator, phenol red, was used to probe the pH change (Supplementary Fig. 48). As shown in Supplementary Fig. 49, we did not observe the change of bulk pH. This result indicates that the activity boost comes from the increase of microenvironmental pH induced by PEI confinement. In addition, we also excluded the potential ability of free PEI to catalyse the hydrolysis of pNPP (Supplementary Fig. 50).

To explore the nanozyme scope, four other typical Zr-MOF nanozymes were modulated by PEI, including UiO-66 NPs, NH2-UiO-66 NPs, UiO-67 NPs, and NU-1000 NPs. After PEI confinement, the obtained nanozymes are designated as UiO-66 NPs@PEI/PEGDE NPs, NH2-UiO-66@PEI/PEGDE NPs, UiO-67@PEI/PEGDE NPs, and NU-1000@PEI/PEGDE NPs (Supplementary Fig. 51). The activity improvement is also achieved (Fig. 6f). The changes of bulk pH of those Zr-MOF nanozymes after PEI confinement are also excluded (Supplementary Fig. 52). We further confined PEI within PCN-222 NPs. It is shown that

PEI is able to increase the fluorescence intensity of PCN-222 NPs, which proves the alkaline microenvironment achieved by PEI confinement. Thus, PEI confinement enhances the hydrolase-like activity of PCN-222 NPs (Supplementary Figs. 53–55). Together, these results convincingly demonstrate that the microenvironmental pH of MOF nanozymes could be increased by introducing polybases.

In summary, we have developed a microenvironmental modulation strategy to break the intrinsic pH limitations of nanozymes. The confinement of polyacid PAA within the channels of PCN-222-Fe nanozymes is able to lower the local pH near the catalytic active centres. MD simulations demonstrate that protons could be trapped within PAA-modified channels to enhance catalytic activity under physiological conditions, a process that is due to the buffering effect of the titratable PAA. The peroxidase-like activity of PCN-222-Fe nanozymes is increased significantly at physiological pH, which further addresses the pH mismatch between natural oxidases and PCN-222-Fe nanozymes. Several analytes have been sensitively detected and chirality recognition of several natural and non-natural amino acids has been achieved based on one-pot oxidases/PCN-222-Fe@PAA cascade reactions. In addition, the confinement of polybase PEI in the cavities or channels of Zr-MOF nanozymes could increase the microenvironmental pH, leading to a significant

improvement in their hydrolase-like activity. We speculate that this strategy will broaden our view into the microenvironment modulation around the catalytic active sites of nanocatalysts.

## Methods

### Fabrication of PCN-222-Fe NPs and PCN-222-Fe@PAA NPs

Typically, 100 mg Fe-TCPP, 300 mg $ZrOCl_2 \cdot 8H_2O$ and 3.3 g BA were dissolved in 100 mL of DMF using a 250-mL flask. The solution was transferred into an oil bath and heated at 90 °C for 5 h with gentle stirring. After cooling to room temperature, the PCN-222-Fe NPs were obtained by centrifugation at $14,800 \times g$ for 20 min and washed with DMF three times. The obtained PCN-222-Fe NPs were stored at 4 °C for further use.

PCN-222-Fe@PAA NPs were obtained by mixing equal volumes of 0.5 mg/mL PCN-222-Fe NPs and 1 mg/mL PAA together and incubating for 30 min at room temperature. The final concentrations of PCN-222-Fe NPs and PAA were 0.25 mg/mL and 0.5 mg/mL respectively. For the activity measurement, the mixture was directly used after incubation. There was no activity difference before and after the removal of free PAA via centrifugation (Supplementary Fig. 17). For the characterisations, such as SEM, TEM, DLS, zeta potential and XPS spectra, PCN-222-Fe@PAA NPs were washed with water three times.

### Peroxidase-like activity measurements

In a typical procedure, equal volumes of 0.5 mg/mL PCN-222-Fe NPs and 1 mg/mL polymers (PAA, PSS, and PAH) were mixed together and incubated for 30 min at room temperature. Forty microliters of the above solution and 40 μL of 10 mM TMB were added into 1.88 mL of 0.2 M Tris buffer (pH = 7.4), followed by the introduction of 40 μL of 10 mM $H_2O_2$. TMB used in the study was in the form of hydrochloride, which was directly dissolved in water. For the fluorescent substrate Amplex Red, 10 μL of 20 mM Amplex Red (in DMSO) and 20 μL of 0.5 mg/mL PCN-222-Fe@PAA NPs were added to 1.95 mL of 0.2 M Tris buffer (pH = 7.4), followed by the addition of 20 μL of 10 mM $H_2O_2$. The absorbance changes of the reaction solutions at 652 nm for TMB and 571 nm for Amplex Red were monitored using a spectrophotometer. The peroxidation of Amplex Red was also monitored by the fluorescent intensity change at 585 nm with an excitation wavelength at 560 nm using a microplate reader. The initial velocities were calculated based on the Beer−Lambert Law.

To investigate the pH dependence on the peroxidase-like activity of PCN-222-Fe NPs or PCN-222-Fe@PAA NPs, 40 μL of 0.25 mg/mL PCN-222-Fe NPs (or PCN-222-Fe@PAA NPs), 40 μL of 10 mM TMB, and 40 μL of 10 mM $H_2O_2$ were added sequentially to 1.88 mL of 0.2 M buffer solution with different pHs (3−6 for acetate buffer, 7−8 for Tris buffer). The absorbance changes at 652 nm were monitored and the initial velocities were calculated.

### Model construction

Starting with the hexagonal crystal system Zr-MOF (deposition number in CCDC: 893545), we built a 5-layer MOF with one PAA chain in each layer and placed it into a periodic water box. Two model systems were further constructed, denoted as 27/0 (all 27 carboxyl groups on a PAA chain are deprotonated, representing a neutral or slightly basic system) and 22/5 (22 carboxyl groups are deprotonated and the other 5 are in protonated states, with the protonation site on each chain randomly generated, representing a weakly acidic system), respectively. Both model systems 27/0 and 22/5 have dimensions of 83.8 * 72.5 * 124.1 Å³. 0.1 M Tris buffer was added to neutralise the system and to mimic the experimental conditions. The total number of atoms is 59,429 in 22/5 and 59904 in 27/0 systems.

### MD simulations and free energy calculations

All atomistic MD simulations were performed with the LAMMPS package[50,51]. The water molecules were described by the TIP3P model, and the SHAKE algorithm was employed to maintain the rigidity of the water molecules[52]. The hydronium was also characterised by the rigidity model. The MOF system was described by the generic universal force field (UFF)[53]. For nonbonding interactions, the cut-off distance was set to be 12.5 Å. The long-range electrostatic interactions were calculated by using the particle mesh Ewald method[54]. In each simulation, energy minimisation was performed, followed by a 150 ps MD simulation using the NPT ensemble (300 K and 1 atmosphere pressure) to equilibrate the density of the systems. Then, 60 ns NVT simulations were carried out at 300 K, and the last 45 ns (for 22/5) or 30 ns (for 27/0) trajectories were used for analysis. The Nosé−Hoover thermostat and barostat algorithm were used to control the temperature and pressure in the simulations. The timestep was set as 0.5 fs, and the trajectories were collected every 100 steps. The free energy calculations were performed using umbrella sampling[55], with a bin size of 3 Å. Sampling in each bin was accumulated for ~3 ns to ensure the convergence of the calculations. The weighted histogram analysis method (WHAM) was used to generate the free energy profile[56].

### Ab initio calculations

We built a model to mimic the local interactions of PAA with hydronium, Tris, or oxTMB. It is a fragment of the PAA chain that contains five carboxyl groups, four of which are deprotonated and one is protonated. Geometry optimisations were performed with the density functional theory (DFT) method at the B3LYP-D3/6-31G* level in the solution phase[57−61]. The polarisable continuum model (PCM) was used to account for the solvation effect with water as the solvent that was used in experiments[62,63]. Vibrational frequencies were calculated for each species to confirm that it is an equilibrium structure (without imaginary frequency). We further performed single-point calculations based on the optimised structures at the B3LYP-D3/6-311++G** level. The Gibbs energy in aqueous solution was corrected to 298.15 K. Thermodynamic correction obtained from frequency analysis was added by solution-phase single-point energy and an additional 1.9 kcal/mol correction was applied to obtain the solution-phase Gibbs free energy used for mechanistic discussion[64]. All of these calculations were carried out with the Gaussian 16 program (Revision A.03)[65].

### Reporting summary

Further information on research design is available in the Nature Portfolio Reporting Summary linked to this article.

## Data availability

The data generated in this study are provided in this paper, its Supporting Information file, and Source Data. Source data are provided with this paper.

## Code availability

The LAMMPS package (Version 23Jun22) can be accessed at https://www.lammps.org/download.html. The Gaussian 16 program (Revision A.03) can be ordered at https://gaussian.com/orders/. VMD (Version 1.9.4) can be accessed at https://www.ks.uiuc.edu/Research/vmd/. The computer code (LAMMPS configuration files, data files, Gaussian input files) along with the analysis scripts used to generate the data in Fig. 4 are available at https://github.com/Xiaoyu1Wang/MD-simulator-and-analyzer[66]. The readers can reproduce the results from this paper using the files uploaded in the link above.

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

## Acknowledgements

This work was funded by the National Natural Science Foundation of China (22374071) (H.W.), 22273034 (H.D.), and 22361142831 (H.D.), the National Key R&D Program of China (2021YFF1200700 and 2019YFA0709200) (H.W.), the National Key R&D Program of China (2023YB3813001) (H.D.), Jiangsu Provincial Key R&D Program (BE2022836) (H.W.) and BF2024056 (H.D.), the PAPD Program (H.W.), the Fundamental Research Funds for the Central Universities (202200325, 021314380228, 021314380195, 0419-14915231) (H.W.), the Sakura Science Program of Japan Science and Technology Agency (JST) (H.W., K.O., S.K.), State Key Laboratory of Analytical Chemistry for Life Science (5431ZZXM2306) (H.W.), the Frontiers Science Centre for Critical Earth Material Cycling of Nanjing University (H.D.), Hunan Provincial Science and Technology Department (2022SK2003 and 2022JJ10007) (H.X.), and Open Funds of NMPA Key Laboratory for Biomedical Optics (20240001) (H.W.). Parts of the calculations were performed using computational resources on an IBM Blade cluster system from the High-Performance Computing Centre (HPCC) of Nanjing University.

## Author contributions

H.W. and H.D. conceived the research. T.L. fabricated the materials, performed the catalysis study, analysed the data and wrote the relevant part of the manuscript. X.W. performed the MD simulations, analysed the data and wrote the relevant part of the manuscript. Y.W., Y.Z., J.W., S.R.L., W.L., S.J.L., Y.L., H.X., K.O. and S.K. assisted with materials characterisations and discussion. The manuscript was revised with contributions from all authors.

## Competing interests

The authors declare no competing interests.
