## [Transparent Peer Review file · Nature Communications]

Microenvironmental modulation breaks intrinsic pH limitations of nanozymes to boost their activities

Corresponding Author: Professor Hui Wei

Version 0:

Reviewer comments:

Reviewer #1

(Remarks to the Author)

This is a nice work on modulating the microenvironment of MOF nanozymes to enhance catalytic performance. The modulation process with PAA and PEI is similar to the pH regulation mechanism using biomolecules in enzyme systems, making it highly valuable. And one-pot oxidase/MOF nanozyme cascade reactions enabled by microenvironmental regulation are significant. I noticed that the authors have effectively addressed most of the previous reviewers' concerns, and I recommend publication after minor revisions.

Detailed comments:

1. Was the same media used for testing Zeta potentials as in the catalytic experiments?
2. There are reports (doi: 10.1002/anie.202202207) on using PEI to regulate pH to enhance activities similar to PTE-like MOF catalysis. Those works are based on PEI encapsulation on the exterior of the MOF, which differs significantly from the catalyst design in this study. Could the authors briefly introduce the impact of structural differences on catalytic performance?
3. Can PAA and PEI be considered consumable reagents? Does this impact the working capacity or cyclability of the catalyst system?
4. Would the doping of PAA and PEI affect the capping agent (organic acids, etc.) on the Zr6 node? Would this influence the nature and activity of the nanozyme? Was any TCPP ligand leaching observed during the doping process, especially for PEI doping?
5. In Fig. 6, the MOF-808 shows the best hydrolase-like activities. Farha's former works about the Lewis acidic open metal sites on different Zr-MOF nodes are good references here to discuss the superiority of this MOF.
6. When discussing the successful confinement of PEI within the cavities of MOF, MOF pore size decrease after loading should be a good indicator.

Reviewer #2

(Remarks to the Author)

In the paper the authors suggest quite an original approach of tuning nanozymes activity, synthesizing composites of catalytically active material with acidic polymers. The latter allowed the authors to shift the optimal working range of nanozymes to physiological pH region. Despite the research does not represent any breakthrough results, matching the status of the Journal, the comprehensiveness of the materials science investigation carried out is impressive. The authors dwell on the proof of the polymer incorporation into the MOF lattice and explain that the observed effect is due microenvironmental pH changes. Unfortunately, the effect is not dramatic – the catalytic activity was only 4-fold increased. Taking into account some interesting findings, I can recommend the article for publication after major revision.

Comments and questions:

1. According to pore size distribution data (Sup. Figure 5), since the pore average size is the same for the composite particles and only their number is decreased, the polymer probably does not incorporate into the bulk of the particles, while blocking the pores in the outer layer of the nanozyme. Firstly, it contradicts the schemes and explanations given. Secondly, whether it blocks the transport of H₂O₂ and reducing substrate into the bulk of the nanozyme? Does the reaction occur only at the nanozyme surface?
2. The authors emphasize that the most of nanozymes display their optimal catalytic activity in acidic media. Probably, this is

due to higher oxidation probability of TMB substrate that is generally used in the access of protons or due to hydrolysis of iron-containing nanomaterials at high pH values. Could you please:

- demonstrate the effect using any substrate with pH-independent redox activity, for example, ferrocyanide,
- comment it in terms of pKa values,
- consider pH-dependent operational stability of the nanozymes described in the paper.

3. 'Due to the pH mismatch, people usually perform the oxidase-catalysed reaction first under neutral conditions, and then the reaction solution is transferred into an acidic media for subsequent reaction catalysed by peroxidase-mimicking nanozymes' – this is not true. Such principle is generally used for the first-generation electrochemical biosensors based on GOD and electrocatalysts of H₂O₂ reduction and is widely applied from early 90s even in physiological pH. Moreover, the whole cycle is known to be limited by glucose mass-transport rather than activity of H₂O₂ reduction catalyst. Accordingly, the dramatic increase in oxidase-nanozyme couple activity (Figure 5e) is rather due to low enzyme activity at pH 4. In any case, how can you explain, such an increase, taking into account that the composite nanozyme activity at pH 7.4 is only 4 times higher than at pH 4?

4. Could you please reduce the number of figures in the palettes. Most of them are useless (for example, the diagrams with zeta potential) or seem to be intermediate - it is inappropriate to put them on the same level with more important results.

Version 1:

Reviewer comments:

Reviewer #1

(Remarks to the Author)

The authors have made significant modifications to address the concerns, and I recommend publication.

Reviewer #2

(Remarks to the Author)

The corrected version can be accepted for publication.

REVIEWER COMMENTS

Reviewer #1 (Remarks to the Author):

This is a nice work on modulating the microenvironment of MOF nanozymes to enhance catalytic performance. The modulation process with PAA and PEI is similar to the pH regulation mechanism using biomolecules in enzyme systems, making it highly valuable. And one-pot oxidase/MOF nanozyme cascade reactions enabled by microenvironmental regulation are significant. I noticed that the authors have effectively addressed most of the previous reviewers' concerns, and I recommend publication after minor revisions.

Reply: We thank the reviewer for the positive comments. We have made the revisions accordingly. Please see the details below.

Detailed comments:

1. Was the same media used for testing Zeta potentials as in the catalytic experiments?

Reply: We thank the reviewer for the question. The media was not the same. The media used in zeta potentials measurements was water for all the MOF NPs. We observed a zeta potential reversal of PCN-222-Fe NPs before and after PAA modification, which indicated the successful modification of PAA. The media used in catalytic experiments was 0.2 M Tris buffer with a pH of 7.4, which maintained the neutral pH of the bulk reaction solution. We have added the media information about zeta potentials tests in the Characterizations section.

2. There are reports (doi: 10.1002/anie.202202207) on using PEI to regulate pH to enhance activities similar to PTE-like MOF catalysis. Those works are based on PEI encapsulation on the exterior of the MOF, which differs significantly from the catalyst design in this study. Could the authors briefly introduce the impact of structural differences on catalytic performance?

Reply: We thank the reviewer for the question. Previous studies have pioneered the effectiveness of PEI as a base for promoting the hydrolase-mimicking activities of Zr-

R
P
A
G
E

MOFs^{1, 2}. Those studies mainly focused on construction of the bulk PEI hydrogels (as cited by the reviewer) or the employment of bulk PEI solution, both of which resulted in an alkaline bulk pH. In our study, we focused on the role of chemical microenvironment in activity enhancement of MOFs. We confined PEI locally on and also within MOFs, which only increased the local pH around the active sites without changing the bulk pH in reaction solutions. In a similar way, the confined PEI was also able to significantly improve the activity of MOFs, which offered us an insight into the power of microenvironment modulation on the enzyme-mimicking activities of MOFs.

3. Can PAA and PEI be considered consumable reagents? Does this impact the working capacity or cyclability of the catalyst system?

Reply: We thank the reviewer for the question. We speculated that PAA and PEI are consumable reagents. Deprotonated PAA, termed PAANa, is not able to improve the peroxidase-mimicking activity of PCN-222-Fe NPs (**Fig. R1a**). The result indicated that proton donation by PAA is essential for the activity enhancement of PCN-222-Fe NPs. Considering that the peroxidation of reducing substrates consumes protons, the activity of PCN-222-Fe@PAA NPs decays with catalysis cycles (**Fig. R1b**). In spite of the activity decrease with catalysis cycles, PAA confinement still guarantees the high cascade efficiency coupled with oxidases and the effective chirality sensing of amino acids during a single use.

Fig. R1: (a) Influence of modification with PAA and deprotonated PAA (*i.e.*, PAANa) on the peroxidase-like activity of PCN-222-Fe NPs. (b) Influence of the catalysis cycle on the peroxidase-like activity of PCN-222-Fe@PAA NPs. Data are expressed as mean \pm standard error of 3 experiments.

As for PEI, we speculated that protonated PEI was not able to significantly improve the hydrolase-mimicking activity of MOF-808 NPs. To prove this, we incubated MOF-808@PEI/PEGDE NPs in HCl solution with a pH of 3.0 for 5 min to protonate the amino groups. After incubation, we immediately removed the HCl solution and washed the NPs with water once through centrifugation. As shown in **Fig. R2a**, PEI modification was able to enhance the hydrolase-mimicking activity by about 9 times. After HCl treatment, we did not observe any considerable increase in the activity of MOF-808 NPs. Besides, the activity of MOF-808@PEI/PEGDE NPs decayed with the catalysis cycles (**Fig. R2b**). We reasoned that the produced phosphate ions after hydrolysis could occupy and poison the catalytically active sites (*i.e.*, Zr₆ clusters) in MOF NPs, which caused the activity decrease with catalysis cycles³.

Fig. R2: (a) Influence of PEI modification and following HCl incubation on the hydrolase-like activity of MOF-808 NPs. The pH of HCl was 3.0, and the incubation time was 5 min. (b) Influence of the catalysis cycle on the hydrolase-like activity of MOF-808@PEI/PEGDE NPs. Data are expressed as mean \pm standard error of 3 experiments.

4. Would the doping of PAA and PEI affect the capping agent (organic acids, etc.) on the Zr₆ node? Would this influence the nature and activity of the nanozyme? Was any TCPP ligand leaching observed during the doping process, especially for PEI doping?

Reply: We thank the reviewer for the question. Abundant Fe-TCPP and a trace amount of the modulator, *i.e.*, benzoic acid, coordinate with the Zr₆ nodes in PCN-222-Fe NPs. Considering that Fe-TCPP serves as the catalytically active site, it is necessary to investigate the influence of PAA incubation on the release of Fe-TCPP. Hence, we collected the supernatant of PCN-222-Fe NPs after PAA incubation, and adjusted the

pH to 7.0 using a phosphate buffer. Free Fe-TCPP was also dissolved in phosphate buffer at pH 7.0 for comparison. It was observed that no Fe-TCPP was released. Not only did PCN-222-Fe NPs, but PCN-222 NPs also showed no release of TCPP after PAA incubation (**Fig. R3**). In view of the results, we excluded that the release of organic linkers after PAA incubation resulted in an activity improvement of MOF NPs.

Fig. R3: (a) UV-vis spectra of 0.25 mg/mL PAA, 5 $\mu\text{g/mL}$ Fe-TCPP, and the supernatant of 0.25 mg/mL PCN-222-Fe NPs after incubation with 0.5 mg/mL PAA. (b) UV-vis spectra of 0.25 mg/mL PAA, 5 $\mu\text{g/mL}$ TCPP, and the supernatant of 0.25 mg/mL PCN-222 NPs after incubation with 0.5 mg/mL PAA. PAA, Fe-TCPP, and TCPP were dissolved in 100 mM phosphate buffer at pH 7.0. The supernatant was added into an equal volume of 200 mM phosphate buffer at pH 7.0 before UV-vis measurements.

Fig. R4: (a) UV-vis spectra of 20 mg/mL PEI, 5 $\mu\text{g/mL}$ TCPP, and the diluted supernatant of 10 mg/mL PCN-222 NPs after incubation with 20 mg/mL PEI. (b) Standard curve of TCPP in ethanol containing 20 mg/mL PEI. The leached TCPP was calculated to be 16.15 $\mu\text{g/mL}$ according to the standard curve.

We further incubated PCN-222 NPs in PEI to monitor the possible release of TCPP ligand. We observed an absorbance signal of TCPP in the supernatant after PEI incubation, which proved the release of TCPP (**Fig. R4a**). To determine the

concentration of the released TCP, we plotted a standard curve of TCP in PEI solution. The leached TCP was calculated to be about 16.15 $\mu\text{g}/\text{mL}$ (**Fig. R4b**). It was noted that the employed concentration of PCN-222 NPs was 10 mg/mL , and the overall TCP concentration in PCN-222 NPs was about 6.58 mg/mL . Hence, the released TCP accounted for only 0.25% of the overall TCP in PCN-222 NPs.

We further monitored the hydrolase-mimicking activity of PCN-222 NPs before and after PEI modification, observing a significant activity improvement following PEI modification (**Fig. R5**). Therefore, we excluded that the release of 0.25% TCP resulted in the activity enhancement of PCN-222 NPs after PEI incubation. We further found that PEI modification did not change the bulk pH, as indicated by the basic pH indicator phenol red (**Fig. R6a**). Instead, we observed an increase in the fluorescence intensity of PCN-222 NPs after PEI modification, which demonstrated that PEI increased the local pH within PCN-222 NPs (**Fig. R6b-c**). The activity enhancement of PCN-222 NPs through PEI modification could be ascribed to the alkaline microenvironment.

Fig. R5: (a) UV-vis spectra of bis(4-nitrophenyl) phosphate (*bNPP*) solutions before and after 20-min incubation with PCN-222 NPs, and PCN-222@PEI/PEGDE NPs. (b) Absorbance changes at 407 nm ($\Delta A_{407 \text{ nm}}$) corresponding to panel a. Data are expressed as mean \pm standard error of 3 experiments.

Fig. R6: (a) UV-vis spectra of 10 μg phenol red (a basic pH indicator) in 1.6 mL of the supernatant of PCN-222 NPs and PCN-222@PEI/PEGDE NPs. (b) Fluorescence emission spectra of 20 $\mu\text{g}/\text{mL}$ PCN-222 NPs and PCN-222@PEI/PEGDE NPs in water. $\lambda_{\text{ex}} = 420$ nm. (c) Normalised fluorescence intensities at 660 nm corresponding to panel b. Data are expressed as mean \pm standard error of 3 experiments.

5. In Fig. 6, the MOF-808 shows the best hydrolase-like activities. Farha's former works about the Lewis acidic open metal sites on different Zr-MOF nodes are good references here to discuss the superiority of this MOF.

Reply: We thank the reviewer for the advice. We have complemented the corresponding discussion in the manuscript. MOF-808 NPs possess 1.8 nm pores, which enable the substrates to access the metal sites inside the NPs. Besides, the Zr_6 nodes in MOF-808 NPs are six-connected, which expose more catalytically active sites than other Zr-MOF NPs, including 12-connected UiO-66 NPs and 8-connected NU-1000 NPs⁴. Hence, we employed MOF-808 as our study model.

6. When discussing the successful confinement of PEI within the cavities of MOF, MOF pore size decrease after loading should be a good indicator.

Reply: We thank the reviewer for the advice. We have revised the discussion accordingly in the manuscript.

Reviewer #2 (Remarks to the Author):

In the paper the authors suggest quite an original approach of tuning nanozymes activity, synthesizing composites of catalytically active material with acidic polymers. The latter allowed the authors to shift the optimal working range of nanozymes to physiological pH region. Despite the research does not represent any breakthrough results, matching the status of the Journal, the comprehensiveness of the materials science investigation carried out is impressive. The authors dwell on the proof of the polymer incorporation into the MOF lattice and explain that the observed effect is due microenvironmental pH changes. Unfortunately, the effect is not dramatic – the catalytic activity was only 4-fold increased. Taking into account some interesting findings, I can recommend the article for publication after major revision.

Reply: We thank the reviewer for pointing out the strengths and weaknesses of our study. We have made the revisions accordingly. Please see the details below.

Comments and questions:

1. According to pore size distribution data (Sup. Figure 5), since the pore average size is the same for the composite particles and only their number is decreased, the polymer probably does not incorporate into the bulk of the particles, while blocking the pores in the outer layer of the nanozyme. Firstly, it contradicts the schemes and explanations given. Secondly, whether it blocks the transport of H₂O₂ and reducing substrate into the bulk of the nanozyme? Does the reaction occur only at the nanozyme surface?

Reply: We thank the reviewer for the questions. According to the pore width distribution result (Fig. R7), there was a slight decrease in the width of the mesopores after PAA confinement. The slight decrease in the pore size was probably due to the flexibility of confined PAA chains. A similar result was observed for peptide chains confined within mesoporous MOFs⁵. To further confirm the confinement of PAA within the mesopores, we performed both PAA concentration-dependent and PAA molecular weight-dependent studies.

R
P
A
G
E

Fig. R7: Pore width distributions of PCN-222-Fe NPs and PCN-222-Fe@PAA NPs.

Fig. R8: Influence of PAA concentration on the microenvironmental pH decrease. (a) Fluorescence emission spectra of 20 $\mu\text{g/mL}$ PCN-222 NPs before and after modification with PAA in 200 mM Tris buffer, $\text{pH} = 7.4$. $\lambda_{\text{ex}} = 420$ nm. During modification, the concentration of PCN-222 NPs was 0.5 mg/mL. (b) Normalised fluorescence intensities at 660 nm corresponding to panel a. (c) Time evolution of $A_{652 \text{ nm}}$ for monitoring the peroxidase-like catalytic activities of 5 $\mu\text{g/mL}$ PCN-222-Fe NPs before and after modification with PAA at R.T., under the condition of 200 mM Tris buffer ($\text{pH} 7.4$) containing 0.2 mM TMB and 0.2 mM H_2O_2 . During modification, the concentration of PCN-222-Fe NPs was 0.25 mg/mL. (d) Initial velocities corresponding to panel c. Data are expressed as mean \pm standard error of 3 experiments.

As mentioned in our original manuscript, the fluorescence intensity of PCN-222 NPs is intrinsically responsive to pH. We previously showed that PAA confinement led to a decrease in the fluorescence intensity of PCN-222 NPs, indicating a local acidic

environment. Here, we lowered the PAA concentration during the modification process, enabling PAA to be primarily adsorbed on the surface of NPs. As shown in **Fig. R8a-b**, a low concentration of PAA was not able to significantly decrease the fluorescence intensity of PCN-222 NPs, which indicated that microenvironmental pH was not lowered effectively. As a result, a low concentration of PAA did not enhance the peroxidase-like activity of PCN-222-Fe NPs significantly (**Fig. R8c-d**). Zeta potential results confirmed that low concentrations of PAA were capable of modifying the surface of MOF NPs (**Fig. R9**).

Fig. R9: (a) Zeta potentials of 0.5 mg/mL PCN-222 NPs before and after modification with 0.1 mg/mL PAA and 1.0 mg/mL PAA. (b) Zeta potentials of 0.25 mg/mL PCN-222-Fe NPs before and after modification with 0.05 mg/mL PAA and 0.5 mg/mL PAA. Data are expressed as mean \pm standard error of 3 experiments.

Furthermore, we employed PAA with a quite high molecular weight of 450 kDa (PAA_{450 kDa}) to modify MOF NPs. In contrast, the PAA confined within MOF NPs possessed a molecular weight of only 2 kDa. It was reasoned that PAA_{450 kDa} would be adsorbed on the surface of MOF NPs. As shown in **Fig. R10a-b**, PAA_{450 kDa} modification hardly lowered the fluorescence intensity of PCN-222 NPs. This proved that PAA_{450 kDa} modification did not create a local acidic environment within PCN-222 NPs. Hence, PAA_{450 kDa} modification was not able to boost the peroxidase-like activity of PCN-222-Fe NPs (**Fig. R10c-d**). Zeta potential results proved the successful modification of PAA_{450 kDa} on the surface of MOF NPs (**Fig. R11**). Taken together, we have shown that PAA chains on the surface of MOF NPs were not able to create an acidic microenvironment. Only those PAA chains confined within the pores lowered the local pH, thus achieving an activity enhancement of PCN-222-Fe NPs under neutral conditions. Therefore, PAA is supposed to be confined within pores in MOF NPs, and

the catalytic sites within pores remain active during reactions.

Fig. R10: Influence of PAA molecular weight on the microenvironmental pH decrease. (a) Fluorescence emission spectra of 20 µg/mL PCN-222 NPs before and after modification with PAA and PAA_{450 kDa} in 200 mM Tris buffer, pH = 7.4. $\lambda_{\text{ex}} = 420$ nm. During modification, the concentration of PCN-222 NPs was 0.5 mg/mL, and the concentrations of PAA and PAA_{450 kDa} were 1.0 mg/mL. (b) Normalised fluorescence intensities at 660 nm corresponding to panel a. (c) Time evolution of A_{652 nm} for monitoring the peroxidase-like catalytic activities of 5 µg/mL PCN-222-Fe NPs before and after modification with PAA and PAA_{450 kDa} at R.T., under the condition of 200 mM Tris buffer (pH 7.4) containing 0.2 mM TMB and 0.2 mM H₂O₂. During modification, the concentration of PCN-222-Fe NPs was 0.25 mg/mL, and the concentrations of PAA and PAA_{450 kDa} were 0.5 mg/mL. (d) Initial velocities corresponding to panel c. Data are expressed as mean \pm standard error of 3 experiments.

Fig. R11: Zeta potentials of (a) PCN-222 NPs, and (b) PCN-222-Fe NPs before and after modification with PAA and PAA_{450 kDa}. Data are expressed as mean \pm standard error of 3 experiments.

PAA indeed restrains the diffusion of substrates towards the catalytic site. We

employed PEG with carboxyl end groups to modify PCN-222-Fe NPs. The molecular weight of PEG was identical to that of PAA. It was shown that PEG decreased the peroxidase-like activity of PCN-222-Fe NPs at both pHs 4.0 and 7.4, probably due to physical blocking (**Fig. R12**). Similar to PEG modification, PAA modification at pH 4.0 also decreased the peroxidase-like activity of PCN-222-Fe NPs at pH 4.0. In contrast, PAA modification at pH 7.4 boosted the activity of PCN-222-Fe NPs, probably due to the protons donated by PAA. It should be noted that PAA at pH 7.4 still restrained the diffusion of substrates to some extent.

Fig. R12: Time evolution of absorbance at 652 nm ($A_{652\text{ nm}}$) for monitoring the peroxidase-mimicking catalytic activities PCN-222 NPs, PCN-222@PEG NPs, and PCN-222-Fe@PAA at (a) pH 4.0 and (b) pH 7.4. (c) Initial velocities corresponding to panels a - b. Data are expressed as mean \pm standard error of 3 experiments. PEG used here is COOH-PEG-COOH with a M_w of 2 kDa.

2. The authors emphasize that the most of nanozymes display their optimal catalytic activity in acidic media. Probably, this is due to higher oxidation probability of TMB substrate that is generally used in the access of protons or due to hydrolysis of iron-containing nanomaterials at high pH values. Could you please:

- demonstrate the effect using any substrate with pH-independent redox activity, for example, ferrocyanide
- comment it in terms of pK_a values,
- consider pH-dependent operational stability of the nanozymes described in the paper.

Reply: We thank the reviewer for the suggestions and have carried out the recommended experiments. First, we investigated the catalytic peroxidation of ferrocyanide using H_2O_2 and our nanozymes. The oxidised product of $K_4Fe(CN)_6$ is $K_3Fe(CN)_6$, so we measured the UV-vis spectra of both $K_4Fe(CN)_6$ and $K_3Fe(CN)_6$ at

different pHs to monitor the reactions (Fig. R13a). The results showed that $\text{K}_3\text{Fe}(\text{CN})_6$ exhibited a characteristic absorption peak at 420 nm, and variations in pH did not influence the absorption coefficient of $\text{K}_3\text{Fe}(\text{CN})_6$ at 420 nm. Therefore, we monitored the absorbance at 420 nm to reflect the production of $\text{K}_3\text{Fe}(\text{CN})_6$. As shown in Fig. R13b, H_2O_2 alone was able to oxidise $\text{K}_4\text{Fe}(\text{CN})_6$. PCN-222-Fe NPs increased the signal of produced $\text{K}_3\text{Fe}(\text{CN})_6$, which indicated the possible catalytic ability of PCN-222-Fe NPs. However, PAA modification did not enhance the peroxidation of $\text{K}_4\text{Fe}(\text{CN})_6$ catalysed by PCN-222-Fe NPs under neutral conditions. In contrast, PAA modification led to a decrease in activity of PCN-222-Fe NPs at all pHs. This was possibly because the redox activity of $\text{K}_4\text{Fe}(\text{CN})_6$ was pH-independent, as indicated by the reviewer.

Fig. R13: Peroxidation of ferrocyanide catalysed by PCN-222-Fe NPs and PCN-222-Fe@PAA NPs. (a) UV-vis spectra of 0.2 mM $\text{K}_4\text{Fe}(\text{CN})_6$ and $\text{K}_3\text{Fe}(\text{CN})_6$ in buffer solutions at different pHs. (b) Optical density at 420 nm ($\text{OD}_{420 \text{ nm}}$) of the supernatants of reaction solutions containing 2 mM $\text{K}_4\text{Fe}(\text{CN})_6$ and 1 mM H_2O_2 without MOF NPs, and with PCN-222-Fe NPs or PCN-222-Fe@PAA NPs after 5-min reaction. The supernatants were obtained by centrifugation. Data are expressed as mean \pm standard error of 3 experiments.

To prove the higher oxidation probability of TMB at lower pHs, we investigated a charge transfer system containing TMB and Ce^{4+} ions. TMB can be oxidised by Ce^{4+} ions, and we studied the influence of pH on this oxidation process. It is known that TMB undergoes both one-electron and two-electron oxidation processes⁶, with the corresponding oxidised TMB products possessing characteristic absorption peaks at 652 nm and 450 nm, respectively (Fig. R14a). Hence, we monitored the absorption signals at both 652 nm and 450 nm in the presence of TMB and Ce^{4+} at different pHs.

As shown in **Fig. R14b**, when the pH decreased from 8.0 to 4.0, the signal at 652 nm increased. This result indicated that the formation of one-electron oxidised TMB increased with the decrease of pH. It could be reasoned that TMB is prone to oxidation under acidic conditions. It was noted that the production of one-electron oxidised TMB decreased when the pH decreased from 4.0 to 3.0. In the meantime, the signal at 450 nm increased obviously, which proved that two-electron oxidised TMB was highly produced at pH 3.0. This result further confirmed the high oxidation probability of TMB at acidic pHs.

Fig. R14: Charge transfer system containing TMB and Ce⁴⁺ ions at different pHs. (a) Schematic illustration of one-electron oxidation and two-electron oxidation of TMB, and its two oxidised TMB species exhibiting different characteristic absorption peaks. (b) Optical density at 652 nm (OD_{652 nm}) and optical density at 450 nm (OD_{450 nm}) of buffer solutions containing 50 μM TMB and 50 μM Ce⁴⁺ ions after 10-min reaction at different pHs. Data are expressed as mean ± standard error of 3 experiments.

We speculated that the protonation of TMB increased its reducibility or oxidation probability. TMB is a typical benzidine compound, possessing two pK_a values of around 3.3 and 4.3⁷. Hence, we depicted the Bjerrum plot of TMB and its protonated species (**Fig. R15**). One TMB molecule adsorbs one proton or two protons, forming TMB-H⁺ or TMB-2H⁺, respectively. It is shown that TMB-2H⁺ dominates at pH 3.0, while few TMB exists. When the pH increases to around 4.5, TMB-2H⁺ essentially disappears and the fractions of TMB-H⁺ and TMB are equivalent. With a further increase of pH to around 6.0, TMB-H⁺ nearly disappears and the fraction of TMB is close to 1.0. When the pH increases to 7.4, TMB is the only species. In short, the amount of protonated TMB species decreases with the increase of pH. The Bjerrum plot corresponds quite well to the pH-dependent results about TMB oxidation by Ce⁴⁺ and TMB peroxidation

by PCN-222-Fe NPs.

Fig. R15: Bjerrum plot of TMB and its protonated species.

We further monitored the pH-dependent stability of PCN-222-Fe@PAA NPs after incubation in both water and buffer solutions at different pHs. As shown in **Fig. R16**, PCN-222-Fe@PAA NPs exhibited decent operational stability over a wide range of pHs.

Fig. R16: pH-dependent stability of PCN-222-Fe@PAA NPs. Relative activity of PCN-222-Fe NPs after 1-h incubation at room temperature in (a) water, and (b) buffer solutions at different pHs. The pH of water in panel a was adjusted by HCl or NaOH. Data are expressed as mean \pm standard error of 3 experiments.

3. ‘Due to the pH mismatch, people usually perform the oxidase-catalysed reaction first under neutral conditions, and then the reaction solution is transferred into an acidic media for subsequent reaction catalysed by peroxidase-mimicking nanozymes’ – this is not true. Such principle is generally used for the first-generation electrochemical biosensors based on GOD and electrocatalysts of H₂O₂ reduction and is widely applied from early 90s even in physiological pH. Moreover, the whole cycle is known to be

limited by glucose mass-transport rather than activity of H₂O₂ reduction catalyst. Accordingly, the dramatic increase in oxidase-nanozyme couple activity (Figure 5e) is rather due to low enzyme activity at pH 4. In any case, how can you explain, such an increase, taking into account that the composite nanozyme activity at pH 7.4 is only 4 times higher than at pH 4?

Reply: We thank the reviewer for the question and the valuable information. We agree that for the first-generation electrochemical biosensors, the overall reaction is limited by the mass transport of an oxidase substrate (such as glucose for a glucose meter). This limitation is probably due to the configuration of electrochemical biosensors, where an oxidase and a H₂O₂ reduction catalyst (such as Prussian blue) are co-immobilized on an electrode. While glucose has to diffuse from the sample to the glucose oxidase on the electrode, H₂O₂ is *in situ* produced and can be catalysed immediately. Consequently, the catalytic glucose oxidation rather than the catalytic H₂O₂ reduction is the limited step in the whole reaction cycle.

In our systems, however, oxidases and peroxidase-like nanozymes are dispersed evenly in the neutral buffer solutions containing substrates, making them easily accessible towards oxidases. Hence, diffusion of the substrates to oxidases is not a limiting process, which is different from that in the electrochemical biosensors. Oxidases such as D-amino acid oxidase (DAAO) exhibit a much higher activity at pH 7.4 than pH 4.0 (**Fig. R17a**). For PCN-222-Fe NPs, their activity is higher at pH 4.0 than pH 7.4. Due to this pH mismatch, the reaction catalysed by DAAO is the rate-limiting step in the cascade reaction at pH 4.0. As DAAO catalysis reaction is the first step, few intermediate H₂O₂ can be produced at pH 4.0, leading to a quite weak signal of the produced TMB_{ox} catalysed by PCN-222-Fe NPs (**Fig. R17b**, line in light gray). When the pH is increased to 7.4, the activity of DAAO is recovered. Although the intermediate H₂O₂ can be produced efficiently, PCN-222-Fe NPs are not able to consume the produced H₂O₂ quickly due to their weak activity at pH 7.4. Nevertheless, PCN-222-Fe NPs still retain the peroxidase-like activity at pH 7.4 to some extent, a

detectable signal of TMBox can be achieved (Fig. R17b, line in gray). In contrast, PCN-222-Fe@PAA NPs possess a 4-5 times higher activity than PCN-222-Fe NPs at pH 7.4, therefore, the signal of the produced TMBox is supposed to be increased by a similar factor (Fig. R17b, line in pink).

Fig. R17: (a) Relative activity of D-amino acid oxidase (DAAO) at pHs 4.0 and 7.4. (b) Time evolution of $A_{652 \text{ nm}}$ for monitoring one-pot cascade reactions catalysed by DAAO coupled with PCN-222-Fe NPs at pHs 4.0 and 7.4, or PCN-222-Fe@PAA NPs at pH 7.4. Data are expressed as mean \pm standard error of 3 experiments.

4. Could you please reduce the number of figures in the palettes. Most of them are useless (for example, the diagrams with zeta potential) or seem to be intermediate - it is inappropriate to put them on the same level with more important results.

Reply: We thank the reviewer for the advice. We have reduced the number of main figures in the manuscript and moved the supporting figures to the SI. After this adjustment, the figures are more concise and focused than original ones (Figs. R18-20).

Fig. R18: Engineering physiological pH active peroxidase-like nanozymes through poly(acrylic acid) confinement. (a) Optimal pHs of peroxidase-like nanozymes reported from 2019 to 2021. (b) Scheme of confining PAA within PCN-222-Fe NPs to form PCN-222-Fe@PAA NPs, leading to an increase in protons near the catalytic active sites and subsequent improvement in the activity of MOF nanozymes. For clarity, the enlarged models were not drawn to scale. SEM images and inset TEM images of (c) PCN-222-Fe NPs and (d) PCN-222-Fe@PAA NPs. (e) pH-dependent initial velocity (v_0) of catalytic peroxidation of TMB, showing the influence of pH on the peroxidase-like activity of PCN-222-Fe NPs and PCN-222-Fe@PAA NPs. (f) Time evolution of absorbance at 652 nm ($A_{652\text{ nm}}$) for monitoring the peroxidase-mimicking catalytic activities of 5 $\mu\text{g/mL}$ PCN-222-Fe NPs or PCN-222-Fe@PAA NPs at R.T., under the condition of 200 mM Tris buffer (pH 7.4) containing 0.2 mM TMB and 0.2 mM H_2O_2 . The inset photos show the colour of the reaction solutions. Data are expressed as mean \pm standard error of 3 experiments.

Fig. R19: Proof of microenvironmental pH decrease through PAA confinement. (a) Scheme of PCN-222 NPs and PCN-222@PAA NPs. The light absorption and fluorescence of PCN-222 NPs are intrinsically responsive to protons concentrations. (b) Photos of 0.5 mg/mL PCN-222 NPs

through PAA modification or introduction of HCl. (c, d) SEM images of PCN-222 NPs and PCN-222@PAA NPs, respectively. (e) $A_{445\text{ nm}}/A_{420\text{ nm}}$ values of 10 $\mu\text{g/mL}$ PCN-222 NPs and PCN-222@PAA NPs in Tris buffer solutions with different concentrations at pH 7.4. (f) Normalised fluorescence intensities at 660 nm of 20 $\mu\text{g/mL}$ PCN-222 NPs and PCN-222@PAA NPs in Tris buffer solutions with different concentrations at pH 7.4. $\lambda_{\text{ex}} = 420\text{ nm}$. Data are expressed as mean \pm standard error of 3 experiments.

Fig. R20: One-pot oxidase/MOF nanozyme cascade reactions enabled by microenvironmental pH modulation. (a) Schematic illustration of pH mismatch between oxidases and peroxidase-mimicking nanozymes, which can be overcome through microenvironmental pH modulation. (b) Schematic illustration of a one-pot cascade reaction catalysed by an oxidase coupled with a peroxidase-mimicking nanozyme, which can be fulfilled through microenvironmental pH modulation. (c) Relative activity of D-amino acid oxidase (DAAO) at pH 4.0 and 7.4. (d) Time evolution of $A_{652\text{ nm}}$ for monitoring one-pot cascade reactions catalysed by DAAO coupled with PCN-222-Fe NPs at pH 4.0 and 7.4, or PCN-222-Fe@PAA NPs at pH 7.4. (e) Schematic illustration of L-amino acid and D-amino acid, which can be recognized through DAAO/MOF nanozyme cascade reaction. (f) Time evolution of $A_{652\text{ nm}}$ for chirality recognition of alanine through DAAO/PCN-222-Fe@PAA cascade reaction. (g) Optical density at 652 nm ($\text{OD}_{652\text{ nm}}$) for chirality recognition of four natural amino acids including alanine, valine, proline, and isoleucine using a microplate reader through DAAO/PCN-222-Fe@PAA cascade reaction. (h) Fluorescence intensity

at 585 nm ($I_{585 \text{ nm}}$) for chirality recognition of natural amino acids as mentioned in panel g. $\lambda_{\text{ex}} = 560$ nm. (i) $I_{585 \text{ nm}}$ for chirality recognition of four non-natural amino acids. Data are expressed as mean \pm standard error of 3 experiments.

References

1. Cheung, Y. H. et al. Environmentally Benign Biosynthesis of Hierarchical MOF/Bacterial Cellulose Composite Sponge for Nerve Agent Protection. *Angew. Chem. Int. Ed.* **61**, e202202207 (2022).
2. Chen, Z., Islamoglu, T. & Farha, O. K. Toward Base Heterogenization: A Zirconium Metal–Organic Framework/Dendrimer or Polymer Mixture for Rapid Hydrolysis of a Nerve-Agent Simulant. *ACS Appl. Nano Mater.* **2**, 1005-1008 (2019).
3. Liao, Y., Sheridan, T., Liu, J., Farha, O. & Hupp, J. Product Inhibition and the Catalytic Destruction of a Nerve Agent Simulant by Zirconium-Based Metal–Organic Frameworks. *ACS Appl. Mater. Interfaces* **13**, 30565-30575 (2021).
4. Moon, S. Y., Liu, Y., Hupp, J. T. & Farha, O. K. Instantaneous Hydrolysis of Nerve-Agent Simulants with a Six-Connected Zirconium-Based Metal–Organic Framework. *Angew. Chem. Int. Ed.* **54**, 6795-6799 (2015).
5. Taketomi, H., Hosono, N. & Uemura, T. Selective Removal of Denatured Proteins Using MOF Nanopores. *J. Am. Chem. Soc.* **146**, 16369-16374 (2024).
6. Zhang, X., Yang, Q., Lang, Y., Jiang, X. & Wu, P. Rationale of 3,3',5,5'-Tetramethylbenzidine as the Chromogenic Substrate in Colorimetric Analysis. *Anal. Chem.* **92**, 12400-12406 (2020).
7. Zierath, D. L., Hassett, J. J., Banwart, W. L., Wood, S. G. & Means, J. C. SORPTION OF BENZIDINE BY SEDIMENTS AND SOILS. *SOIL SCI.* **129**, 277-281 (1980).